# Targeting Neutrophil β_2_-Integrins: A Review of Relevant Resources, Tools, and Methods

**DOI:** 10.3390/biom13060892

**Published:** 2023-05-26

**Authors:** Haleigh E. Conley, M. Katie Sheats

**Affiliations:** 1Department of Clinical Sciences, College of Veterinary Medicine, North Carolina State University, Raleigh, NC 27607, USA; 2Comparative Medicine Institute, North Carolina State University, Raleigh, NC 27607, USA

**Keywords:** neutrophils, β_2_-integrins, CD18, CD11, ICAM-1, innate immune cells

## Abstract

Neutrophils are important innate immune cells that respond during inflammation and infection. These migratory cells utilize β_2_-integrin cell surface receptors to move out of the vasculature into inflamed tissues and to perform various anti-inflammatory responses. Although critical for fighting off infection, neutrophil responses can also become dysregulated and contribute to disease pathophysiology. In order to limit neutrophil-mediated damage, investigators have focused on β_2_-integrins as potential therapeutic targets, but so far these strategies have failed in clinical trials. As the field continues to move forward, a better understanding of β_2_-integrin function and signaling will aid the design of future therapeutics. Here, we provide a detailed review of resources, tools, experimental methods, and in vivo models that have been and will continue to be utilized to investigate the vitally important cell surface receptors, neutrophil β_2_-integrins.

## 1. Introduction

Neutrophils are the predominant circulating leukocytes in the blood and are considered the first responders of the immune system. Neutrophils defend the host against invading pathogens via effector functions such as respiratory burst, phagocytosis, and the release of NETs (see Abbreviations). To accomplish these tasks, neutrophils must travel out of the vasculature and into the injured or infected tissue through a process known as transmigration. β_2_-integrins are specialized cell surface receptors that play a key role in a neutrophil’s ability to transmigrate. The mechanisms of β_2_-integrin function and signaling have been researched and reviewed extensively [1,2]. While much has been learned, therapeutic efforts to target β_2_-integrins to mitigate neutrophil-mediated host injury or disease have not proved clinically beneficial. Because neutrophils play a role in the pathophysiology of diseases ranging from acute lung injury and sepsis to rheumatoid arthritis and organ transplant rejection, the methods used to study neutrophil β_2_-integrins, reviewed here, remain of interest to a wide array of basic, translational, and clinical health researchers.

## 2. β_2_-Integrins and Neutrophils

### 2.1. β_2_-Integrin Activation and Signaling

Expressed exclusively on leukocytes, β_2_-integrins are transmembrane heterodimers that consist of a common β-subunit (CD18), which is non-covalently associated with one of the four known α-subunits (CD11a,b,c,d) (Table 1) [3,4]. The two most prominent and most studied integrins on neutrophils are LFA-1 (α_L_β_2_) and Mac-1 (α_M_β_2_). The α_D_β_2_ integrin is the least researched β_2_-integrin; however, a recent review extensively covers what is known about this integrin [4]. Within circulating, quiescent neutrophils, α_M_β_2_ integrins are primarily contained within the cytoplasm, the secondary and tertiary granules, and the secretory vesicles. On the surface of resting neutrophils, α_L_β_2_-integrins are in an inactive or ‘bent’ conformation, termed the “low-affinity” state. This combination of low surface expression and inactive conformation are control measures to help prevent non-specific neutrophil binding and activation, as unregulated activation could lead to damaging effects for the host. It is only when neutrophils encounter an activation signal, such as the binding of a chemoattractant (e.g., leukotriene B4 (LTB_4_), N-formylmethionine-leucyl-phenylalanine (fMLP)) to a G-protein-coupled receptor (GPCR), that α_M_β_2_-integrin surface expression is increased, conformational changes take place (“affinity”) to open β_2_-integrins, and increased mobility within the membrane leads to cluster formation (“avidity/valency”). This method of activation in which integrin affinity and avidity are altered by intracellular signals that affect change at the integrin cytoplasmic tail is known as “inside-out” activation [5,6]. In contrast to “inside-out,” “outside-in” activation occurs when the β_2_-integrin extracellular domain interacts directly with extracellular matrix proteins or other cell surface ligands (ICAM-1, fibrinogen, etc.) and initiates its own signaling [1,7]. This triggers the phosphorylation of ITAM-bearing transmembrane adapters DAP-12 and Fcγ receptors (FcγRs), which go on to activate Syk and initiate a signaling cascade for cytoskeletal reorganization [8]. These effects on the cytoskeleton are important for the role of β_2_-integrins in neutrophil adhesion strengthening, cell spreading, and crawling [9]. Despite being described, and studied in vitro, as distinct pathways, inside-out and outside-in activation are designed to work in concert in vivo, with signaling from one pathway reinforcing the other, and vice versa.

### 2.2. Neutrophils in Disease

Neutrophils are essential as the “first responders” of the immune system, and without these cells patients are at increased risk from infection or injury. The clearest illustration of this is the frequent and life-threatening infections experienced by patients who lack functional β_2_-integrins due to Leukocyte Adhesion Deficiency (LAD). However, although neutrophils are essential for the maintenance of life and health, they can also cause damage to host tissue in numerous chronic inflammatory conditions and acute inflammatory events, making neutrophil-targeting therapies highly desirable. For example, patients with severe SARS-CoV-2 experience an influx of neutrophils into the lungs, resulting in alveolar damage and the development of acute respiratory distress syndrome (ARDS) [10,11]. Numerous diseases and disorders involve neutrophils and specifically β_2_-integrins (Table 2). Neutrophil β_2_-integrins also bind pathogen-associated molecular patterns (PAMPs), such as LPS and β-glucans [12,13,14,15]. Thus, neutrophils can also become activated directly by pathogens during infection, resulting in effector responses, such as respiratory burst, phagocytosis, and NET formation. Because of the essential role integrins play in neutrophil inflammatory functions, they are attractive therapeutic targets [16,17]. However, β_2_-integrin-targeting therapies have not been successful in clinical trials [18], and additional research is needed to identify new methods of targeting these integrins. Within the scientific literature, there have been many approaches to studying β_2_-integrins. This review article will provide an overview of methods used for investigating β_2_-integrin function, activation, and signaling in neutrophils. Hopefully, with the use of resources, tools and methods presented here, and the continued development of new approaches, researchers will discover a more comprehensive understanding of β_2_-integrins that will lead to successful therapeutic targets to benefit patients with neutrophil-mediated disease. 

## 3. Cell Types and Tools for Evaluating Integrins

### 3.1. Primary Cells

Neutrophils are hematopoietic cells that are terminally differentiated from myeloblasts. They are the most predominant circulating leukocytes in the blood and their typical life span in circulation ranges between less than 24 h to 5.4 days [56,57]. Most primary neutrophil research uses cells collected from either humans or mice. Over the years, several methods have been utilized to isolate neutrophils from human peripheral blood. Most protocols include erythrocyte sedimentation and density centrifugation. Erythrocyte sedimentation is typically achieved using dextran (varying concentrations 1–6%) or HetaSep [58]. For density centrifugation, Ficoll and Percoll are the most commonly utilized options. The order of these steps often varies depending on the research group. Despite their common usage, there is concern that neutrophils isolated by Dextran and Ficoll are prematurely activated by the presence of monocytes [59]. To avoid this background stimulation, some use a one-step high-density Ficoll (1.114 g/mL) without erythrocyte sedimentation, or substitute a discontinuous gradient for Ficoll [59,60]. Neutrophils may also be “rested” following isolation prior to functional assays to reduce unintended activation [61]. Red blood cell lysis usually follows the isolation of neutrophils; however, when the experimental method does not require the removal of red blood cells, such as flow cytometry, avoiding lysis may be one strategy to prevent unwanted neutrophil activation.

These isolation methods routinely yield normal-density neutrophils (NDNs) but fail to isolate the low-density neutrophils (LDNs) that exist in individuals with inflammatory conditions. To isolate LDNs, negative selection by magnetic beads of both the peripheral blood mononuclear cell (PBMC) layer and the granulocyte layer is necessary [58,62]. While the application of magnetic microbeads facilitates the isolation of pure cell populations, this method increases the cost of isolation and may still require lysis to remove contaminating RBCs [63]. Neutrophils isolated via negative selection by microbeads do not display iatrogenic activation because the labeling antibodies are not directed at neutrophils [64]. In fact, magnetic separation of neutrophils results in significantly lower iatrogenic activation compared with traditional dextran sedimentation followed by density centrifugation with Percoll [65,66].

Primary human neutrophils are easy to obtain from willing donors, and a major benefit of using primary human cells is the ability to obtain samples from humans with diseases of interest. Neutrophils from LAD-I patients have been an invaluable resource for researchers interested in β_2_-integrin-dependent and independent neutrophil functions and cell signaling downstream of β_2_-integrins [35,67]. However, the risk to the patient versus the benefit of health discovery research must also be a consideration when obtaining samples from patients. For this reason, volume and cell number are likely to be even more limited when samples are obtained from diseased patients. While sampling from human populations can be extremely convenient, there are logistical considerations, such as Institutional Review Board (IRB) approvals and the need for technically skilled personnel, which often restricts human research with primary neutrophils to experienced labs. This was especially true during the COVID-19 pandemic when IRB protocols changed to reduce the risk for participants and researchers, resulting in reduced access to human participants. Commercially available (e.g., iQ Biosciences^®^, HemaCare^®^) cryopreserved products are a potential alternative for researchers seeking to perform experiments with primary neutrophils; however, while cryopreserved neutrophils retain some phagocytic and migratory functions, these are diminished compared with freshly isolated cells [68]. Additionally, preservation of the oxidative metabolism and microbicidal activity requires specialized storage techniques [69]. 

In addition to primary cells, enucleated neutrophils (cytoplasts) and neutrophil-derived extracellular vesicles also have functional activity. Cytoplasts are enucleated neutrophils that retain very similar chemotactic and bactericidal activity as their parent neutrophils, and these functions are reportedly intact following cryopreservation [70,71]. Neutrophil-derived extracellular vesicles (EVs) also have functional antibacterial activity and express common neutrophil surface receptors, such as CD11b and CD18 [72,73,74]. Interestingly, neutrophil EVs can be stimulated through Mac-1 clustering, highlighting the downstream signaling that occurs following outside-in β_2_-integrin clustering [75]. Although the information on neutrophil EVs is still limited, future studies should interrogate the intracellular signaling related to the β_2_-integrin involvement of these EVs to complement the amassing antimicrobial functional data.

Primary neutrophils from mice are commonly isolated from bone marrow or peripheral blood. Similar to humans, density centrifugation with either Percoll or Histopaque discontinuous gradients is utilized [76,77,78]. Due to the wide availability of species-specific resources, murine neutrophils can also be isolated from bulk populations using magnetic microbeads or by fluorescence-activated cell sorting (FACS) [79,80]. Murine neutrophils are widely used due to the availability of models that duplicate neutrophil function during health and disease. Although they express the same β_2_-integrins, mouse neutrophils do not provide a perfect parallel to humans. In humans, neutrophils are the predominant circulating cell type in the blood (50–70% neutrophils, 30–50% lymphocytes), whereas mice have an abundance of lymphocytes (10–25% neutrophils, 75–90% lymphocytes) [81]. Additionally, murine neutrophils do not express the same FcγRs as human neutrophils [82,83]. Murine neutrophils isolated from bone marrow also display different surface markers and functional activities compared with neutrophils harvested from peripheral blood due to the presence of more immature neutrophils and neutrophil precursors in the bone marrow. Magnetic bead selection is one method that can improve the isolation of mature neutrophils from mouse bone marrow [84,85,86].

Neutrophils are a heterogenous population consisting of normal-density neutrophils (NDNs), low-density neutrophils (LDNs), immature neutrophils, mature neutrophils, and neutrophils with immunosuppressive capabilities [87]. Although mice do express these neutrophil subpopulations, they do not always mirror the presentation of that in humans. During acute infection and inflammation, murine peripheral blood neutrophils exhibit both proinflammatory (CD11b^−^CD49d^+^IL-12^+^) and anti-inflammatory (CD11b^+^CD49d^−^IL-10^+^) neutrophil subsets that have not yet been identified in humans [87]. Human autoimmune disease often results in increased circulation of proinflammatory LDNs, but murine models of autoimmune disease do not display these heterologous populations of neutrophils [87]. These phenotypic differences have been linked with functional differences as well, specifically in murine cancer models [87,88]. Furthermore, in a study conducted by Soroush et al., stimulation of murine pulmonary endothelial cells with tumor necrosis factor α (TNFα) did not result in the upregulation of ICAM-1 expression whereas TNFα stimulation of human pulmonary endothelial cells did induce ICAM-1 upregulation [89]. Thus, it may be more difficult to model β_2_-integrin-dependent interactions of neutrophils with murine-derived endothelial cells. Human neutrophils also have significant transcriptional and epigenetic diversity. Females especially have elevated gene expression levels related to immune responses that correspond with increased occurrences of autoimmune disease [90]. Further, mice cannot model the impact of ethnic diversity on neutrophils, despite recent advances in high-diversity mouse populations [91,92].

### 3.2. Cell Lines

While freshly isolated primary cells are highly desirable for understanding neutrophil β_2_-integrins, they are not always accessible or suitable for certain experiments. For example, although reported [93,94], the manipulation of RNA and protein expression in primary neutrophils is extremely difficult, so cell lines are beneficial for researchers aiming to investigate the roles of individual proteins through knockdown or overexpression studies. Despite some limitations and drawbacks of neutrophil-like cell lines (Table 3), they can be a useful approach for studying integrin function and signaling via induced mutations, rather than having to develop a new transgenic mouse line. 

The HL60 cell line is a human promyeoloblast cell line that can be differentiated into neutrophil-like cells utilizing dimethylsulfoxide (DMSO) or retinoic acid [95,96]. PLB-985 cells are a genetically identical subline of HL60 that are also differentiated using DMSO or retinoic acid [97,98,99]. Both methods of differentiation in HL60s and PLB-985s result in mature neutrophil-like cells; however, compared with DMSO, differentiation with retinoic acid resulted in dampened cellular responses to fMLP and increased random cellular migration [100]. The two methods also result in different expression levels of Scar1 and WASP proteins [101]. Functionally, DMSO-differentiated HL60 neutrophil-like cells are mostly similar to primary neutrophils but do express some differences (Table 3) [95,97,102,103,104]. Despite these differences, research using mutated HL60s has contributed to our understanding of LFA-1 in migration [105]. HL60s have also been used to model host–pathogen interactions [106]. 

Although not as commonly used as HL60s, the human myeloid cell line K562 has been utilized over the past 15 years in research focusing on granulocytes and β_2_-integrins. Xue et al. used the K562 cell line to determine the impacts of kindlin-3 defects on integrin function. Through these studies, it was demonstrated that kindlin-3 is required for β_2_-integrin-mediated adhesion and cell spreading [107]. Another group of researchers used K562 cells as a means to express constructs of α_M_ fused to mCFP and β_2_ fused to mYFP to assess Mac-1 cytoplasmic tail separation during integrin activation via FRET analysis [108]. With this technique, investigators determined that integrin-ligand binding, or integrin crosslinking, induced Mac-1 cytoplasmic tail separation, which was essential for triggering outside-in signaling pathways. This key finding offers insight into why this strategy of leukocyte adhesion blockade failed in clinical trials, as these integrin ligand mimetics designed to block neutrophil-endothelial adhesion were activating neutrophils through a different pathway [109,110]. K562 cells have been further used to evaluate potential small peptide inhibitors and monoclonal antibodies directed against β_2_-integrins [111]. Another benefit of K562 cells is that they only express the transfected β_2_-integrin, allowing for studies examining only Mac-1 or LFA-1, if desired. They also respond similarly to common stimuli of primary neutrophils, including Mn^2+^ [108,112]. 

HoxB8 cells are immortalized murine hematopoietic progenitors that are differentiated into neutrophil-like cells using GM-CSF treatment. These cells perform many neutrophil and integrin-mediated functions similar to primary murine neutrophils, with a few discrepancies (Table 3) [113,114,115,116]. The usage of these cells has increased over the past several years, primarily because HoxB8 cells can be generated from transgenic mice and used to examine specific signaling molecule interactions related to β_2_-integrins [117,118]. They can also be engrafted into naïve mice and functionally respond to bacterial pathogens [116]. Recently, investigators used HoxB8 neutrophil-like cells to show that Rap1 and Riam binding to talin is critical for β_2_-integrin function [117]. In addition to traditional methods of transfection and siRNA, HoxB8 cells can also be manipulated by CRISPR/Cas9 technology to achieve mutants of interest [117,118]. Importantly, murine HoxB8 neutrophil-like cells expressing a human β_2_-integrin ortholog display fully functional signaling and adhesive properties in response to common stimuli, such as PMA, TNFα, etc. [118]

### 3.3. Tools

#### 3.3.1. Anti-Integrin and Fluorescently Labeled Antibodies

Antibodies, including anti-integrin antibodies, are a common tool used to investigate β_2_-integrins [67,106,119,120,121] (Table 4). These antibodies are advantageous for common lab use due to their ease of application and relatively low cost. They are routinely used in three different ways: function blocking, integrin crosslinking, or fluorescent labeling. As a tool to block function, anti-integrin antibodies led to the discovery that Mac-1 is responsible for neutrophil firm adhesion [67]. However, antibody binding of β_2_-integrins can also cause activation, as demonstrated by antibody stimulation of neutrophils in the absence of a ligand (e.g., ICAM-1) and subsequent β_2_-integrin outside-in activation and signaling [108]. This dual nature requires careful experimental planning to prevent unintentional crosslinking and/or Fc receptor engagement when used for inhibitory applications [122]. To avoid these unintentional interactions, researchers can use F(ab) and F(ab)’_2_ fragments derived from monoclonal antibodies [123,124]. These fragments are portions of antibodies where the Fc fragments are cleaved off to prevent non-specific binding of Fc receptors to antibodies. Both types of fragments are helpful in blocking antibodies, and F(ab)’_2_ provide additional capabilities for precipitating proteins of interest. 

The use of fluorescently labeled antibodies has also been an invaluable tool for researchers. Using specific antibodies, surface expression levels of integrins, including the bent versus open conformations, can be examined quantitatively and qualitatively [125]. This methodology led to the discovery that neutrophils from patients with antiphospholipid syndrome (APS) have an upregulation of activated CD11b, which contributes to increased neutrophil adhesiveness [126]. Fluorescently labeled antibodies can also be used in vivo. Wilson et al. administered PE-anti-Ly6G intravenously to evaluate neutrophil infiltration induced by *P. aeruginosa* in talin-1 or kindlin-3 knockout mice [127]. As researchers continue to seek novel protein targets to regulate β_2_-integrins, fluorescently labeled antibodies combined with flow cytometry and/or microscopy may be a first step to understanding the impact inhibitors may have on β_2_-integrin expression and activation. 

**Table 4 biomolecules-13-00892-t004:** Common antibodies used to interrogate β_2_-integrins.

Antibody	Clone	Conformation/Purpose	References
Anti-CD18	IB4	Recognizes CD18 expression	[128,129]
		Crosslinking of CD18	
		In vitro blocking of human β_2_-integrins	
	GAME-46	Recognizes murine CD18 expression In vitro and in vivo blocking of murine CD18	[127,130,131]
	CBR LFA-1/2	Crosslinking of CD18 Recognizes CD18 expression	[105,132]
Anti-CD11b	CBMR1/5	Recognizes high-affinity /activated CD11b	[133,134,135]
Anti-CD11b	ICRF44	Recognizes CD11b expression	[136]
	M1/70	Recognizes CD11b expression In vivo blocking of murine CD11b	[137,138]
Anti-human β_2_-integrin	KIM127	Recognizes bent low-affinity (E^+^H^−^) β_2_-integrin conformation	[135,139]
Anti-human CD11a/CD18	m24	Recognizes extended/high-affinity (H^+^) β_2_-integrin conformation	[139]

#### 3.3.2. Divalent Cations

Divalent cations (e.g., Mn^2+^, Ca^2+^, Mg^2+^) are required for many biological processes, including the binding of integrins to their ligands. Both manganese (Mn^2+^) and magnesium (Mg^2+^) act by binding the metal-ion-dependent adhesion site (MIDAS) domain. Mn^2+^ binding to the MIDAS domain induces outside-in β_2_-integrin activation by forcing integrins to assume a high-affinity conformation that enhances ligand binding [105]. Because of this effect on the MIDAS domain, Mn^2+^ can also be applied as a rescue strategy when examining integrin defects caused by mutation or chemical inhibition. In the absence of inside-out activation signals, Mn^2+^-stimulation can be used as the proximal-most event in the outside-in β_2_-integrin signaling cascade. Using this approach, Xu et al. determined that Mn^2+^ treatment could not rescue the binding defects of myosin light chain kinase (MYLK)-deficient murine neutrophils, indicating a critical role for MYLK in outside-in β_2_-integrin activation [45]. 

Like manganese, calcium and magnesium are required for biological processes. Therefore, researchers commonly include calcium and magnesium supplementation in media, and manipulation of cation presence has led to a better understanding of integrin regulation. Calcium chelation is known to decrease integrin expression and is often used as a positive control for inhibition in experiments [112,134]. Divalent cation stimulation of neutrophils with manganese or higher concentrations of magnesium induces the high-affinity conformation of β_2_-integrins without activating the neutrophil itself or increasing β_2_-integrin surface expression [140]. Through the manipulation of cation concentrations, Spillmann et al. demonstrated that β_2_-integrins must be in their active/high-affinity states to mediate adhesion [140]. A complete understanding of how divalent cations impact neutrophil function and integrin activation is also useful for interpreting clinical information following certain treatments. For example, magnesium sulfate treatment for preterm birth impairs neonatal innate immune cell recruitment and β_2_-integrin-dependent neutrophil responses [141]. 

### 3.4. Common Ligands

#### 3.4.1. Recombinant ICAM-1

β_2_-integrins bind to intercellular adhesion molecules (e.g., ICAM-1) expressed on the surface of endothelial cells to transmigrate from the vasculature into inflamed tissues. This binding interaction induces outside-in activation and signaling of neutrophil β_2_-integrins. One of the most utilized ligands for understanding β_2_-integrin activation and signaling is ICAM-1 because it is a powerful tool for modeling physiologically relevant neutrophil interactions. In vitro, ICAM-1 stimulates neutrophil activation and adhesion in shear flow assays and even induces the clustering of neutrophil β_2_-integrins [33,45,108,112]. Although the usage of recombinant ICAM-1 is extremely common, there have been several discrepancies in the literature surrounding the nomenclature and usage of this ligand. Recombinant ICAM-1/Fc is often used interchangeably with recombinant ICAM-1. Based on our own observations (unpublished findings) and those cited in the literature, the Fc domain of ICAM-1/Fc is likely engaging Fc rectors on neutrophils and causing an inside-out activation cascade [122,133]. In light of this finding, we suggest that the choice of ICAM-1 construct is critical for the appropriate design of experiments interrogating neutrophil inside-out or outside-in activation, or both. 

#### 3.4.2. Fibrinogen

Fibrinogen is a glycoprotein found in the blood that is enzymatically converted to fibrin to promote clotting after damage occurs to vasculature or tissues. Neutrophil β_2_-integrins bind fibrinogen at sites of inflammation; therefore, it is used in vitro to determine integrin-dependent responses [33,113,142,143,144,145]. Lowell et al. determined that Src family kinases were important for β_2_ and β_3_-integrin signaling by evaluating hck^−/−^ fgr^−/−^ double mutant murine neutrophils on fibrinogen [142]. The double mutant neutrophils failed to spread on fibrinogen, but PMA stimulation was able to overcome the defect, indicating that Src kinases act upstream of PKC during integrin-mediated signaling. Fibrinogen initiates outside-in integrin signaling in both β_2_ and β_3_-integrins [145,146]; therefore, experiments utilizing this ligand for β_2_-integrins must rule out effects caused by β_3_-integrin engagement as well. 

#### 3.4.3. PolyRGD

β-integrins bind extracellular matrix proteins (e.g., fibrinogen, fibronectin, collagen, and von Willebrand factor) via their RGD (arginine-glycine-aspartic acid—RGD) site [147]. PolyRGD is a synthetic tripeptide used to engage integrins, and it is known for producing a robust CD18-dependent respiratory burst [54,148]. Other investigations found that Fc receptor knockout mice had decreased respiratory burst in response to polyRGD [149], suggesting that inside-out activation via Fc receptors, or Fc receptor cooperation, may also play a role in neutrophil responses to polyRGD. Because RGD binding sites exist on all β-integrins, polyRGD stimulates β_1_, β_2_, and β_3_-integrins expressed on neutrophils [84,150,151]. Therefore, the PolyRGD may not the best tool for isolating β_2_-integrin activation and signaling. However, it could be a useful tool for investigators interested in redundancy or cross-talk between parallel β-integrin activation and cell signaling cascades.

#### 3.4.4. iC3b

Complement C3 fragment iC3b is a component of the complement system formed when complement factor I cleaves C3b. β_2_-integrins bind iC3b and are recognized as complement receptor 3 (CR3) [152]. In a physiological context, iC3b is an opsonin to support β_2_-integrin-mediated phagocytosis of pathogens [25]. Xue et al. used iC3b to stimulate K562 kindlin-3 knockdown cells and determined that kindlin-3 is required for iC3b-mediated outside-in β_2_-integrin signaling [107]. IC3b can also be used as a coating for neutrophil adhesion or in shear flow experiments [107]. Assays using iC3b as a tool are likely most relevant for in vitro modeling of diseases that may have iC3b-containing immune complex deposition contributing to neutrophil aggregate formation, such as Systemic Lupus Erythematosus (SLE) [12]. 

### 3.5. Assays

#### 3.5.1. Flow Cytometry

Flow cytometry is a high-throughput technology that analyzes single cells from bulk populations. The technology detects and measures physical and chemical characteristics based on cell size and fluorescence. Neutrophils can be easily distinguished using flow cytometry based on their size and high granularity determined by a high side scatter profile when evaluating both side and forward scatter measurements. The usefulness of flow cytometry is widely known across many fields of research, and leukocyte researchers have also harnessed this powerful tool to assess β_2_-integrins. Using fluorescently labeled antibodies, researchers can measure the expression, avidity, and affinity of β_2_-integrins to learn more about how a protein or inhibitor impacts expression or to determine whether certain diseases cause changes in β_2_-integrin expression. Flow cytometry can also be used to measure neutrophil binding to ligands, such as ICAM-1, in the presence of pharmacological inhibitors or when isolated from transgenic mice [153,154]. Flow cytometry analysis of integrin expression is a relatively easy but powerful assay to complement other experiments. One caution is that neutrophil populations may exhibit autofluorescence, including autofluorescence attributed to contaminating eosinophils [85,155,156]. Non-specific staining can also occur if excessive concentrations of labeled antibodies are used, illustrating the importance of concentration optimization [157]. A significant benefit of flow cytometry is that multiple aspects of the neutrophil can be evaluated at once, such as integrin expression and cell viability. Imaging flow cytometry is a newer methodology used in neutrophil research. Specifically, this technology can be used to measure the fluorescence and morphology of neutrophils during functions, such as phagocytosis [158]. Because of its higher power in cellular analyses, this technique provides a breadth of information about cells. However, large data files can create challenges for data management and analysis [159]. 

#### 3.5.2. Static Adhesion

Static adhesion is a common assay that has been used to assess neutrophils for over twenty years. Fluorescently labeled (e.g., calcein AM) neutrophils are added to ligand-covered plates for a designated time followed by subsequent washing and fluorescence readings [123,160]. This assay can evaluate the adhesion of neutrophil and neutrophil-like cells on most ligands, including human umbilical vein endothelial cell (HUVEC) monolayers [161,162]. Unfortunately, static adhesion assays are subject to technical variability due to the inversion procedure to “dump” cells. Further, static adhesion assays cannot fully recapitulate the physiological environment that occurs during shear flow adhesion. For example, neutrophil migration and adhesion under static adhesion require vinculin; however, vinculin was not required for integrin-mediated migration and adhesion when neutrophils were examined under shear flow [115]. Despite these differences, static adhesion assays do offer a high-throughput means to examine β_2_ integrin-mediated firm adhesion [66,123]. 

#### 3.5.3. FRET

Förster Resonance Energy Transfer (FRET) (also referred to as Fluorescence Resonance Energy Transfer) is a method that shows energy transfer between two light-sensitive molecules based on distance [163]. With this newer technology, two proteins of interest can be labeled to quantitatively measure the interactions between the proteins. FRET can detect neutrophil β_2_-integrin conformational changes in the extracellular domain and the cytoplasmic tail when the α and β chains are labeled separately. Lefort et al. used this method to determine how inside-out activation of Mac-1 results in integrin headpiece extension from the bent conformation. Cytoplasmic domain FRET in K562 cells demonstrated that Mac-1 binding to ICAM-1 resulted in the separation of integrin cytoplasmic tails [108]. The sensitivity of this method has made it easier to determine protein interactions within living cells, including how integrins respond during neutrophil stimulation.

#### 3.5.4. Integrin Crosslinking

Integrin crosslinking is a technique where anti-integrin antibodies (e.g., anti-CD18 mAb) are coated on a plate and used as the stimulus for β_2_-integrin activation and signaling [148]. This approach has historically been used to induce outside-in signaling of integrins. However, Jakus and colleagues determined that there was cooperative interaction between FcγRIIa and integrins during integrin crosslinking with anti-CD18 mAb due to the presence of Fc domains on mAbs. In this study, they demonstrated that anti-CD18 mAb crosslinking resulted in neutrophil respiratory burst. When anti-CD18 F(ab’)_2_ was used instead, respiratory burst no longer occurred despite significant neutrophil adhesion. This finding helped to prove that full neutrophil activation resulting in respiratory burst requires more than just integrin-ligand binding and outside-in β_2_-integrin activation [122]. This technique provides many benefits to understanding the cooperative signaling of integrins and FcγRs, but researchers should elect to use F(ab) or F(ab’)_2_ fragments, rather than intact antibodies, when trying to limit their stimulation to outside-in β_2_-integrin activation.

#### 3.5.5. Flow Chamber Assays

Neutrophils are migratory cells where dynamic motion is a critical part of their function. Many in vitro neutrophil assays are unable to capture the dynamic process of neutrophil diapedesis. Flow chamber experiments can determine neutrophil crawling velocity, arrest, polarization, migration patterns, and diapedesis using microscopy. This technique offers a multitude of options for the use of ligands, cell type (whole blood, primary or differentiated neutrophil-like), chemoattractants, function-blocking antibodies, and immunofluorescence microscopy [54,137,164]. Flow chambers can also be coated with desired ligands and perfused with whole blood via tubing directly attached to murine carotid arteries. With this approach, Zarbock and colleagues demonstrated that E-selectin engagement is required for LFA-1-dependent rolling on ICAM-1 [165]. Microfluidic systems are also often used to determine the strength of neutrophil adhesion in relation to the ligand or a known amount of tension [137,166]. Morikis et al. demonstrated that neutrophils had increased adhesion and calcium flux in response to higher tension ligands while under shear flow [166]. Their study highlighted how high-affinity neutrophil β_2_-integrins recognize different levels of shear stress and tension and modulate downstream function and signaling to correspond to the stimulus [166].

In addition to the ligands utilized in shear flow assays, neutrophil interactions with cell monolayers can also be evaluated. Sule and colleagues demonstrated that neutrophils from patients with antiphospholipid syndrome display increased adhesion to HUVECs due to upregulated β_2_-integrin activation [126]. This system can also be designed to model organ-specific neutrophil interactions, such as blood-brain barrier inflammation. Gorina et al. showed that neutrophils use β_2_-integrins to crawl on ICAM-1 prior to diapedesis across isolated primary mouse brain microvascular endothelial cells [23]. In summary, the advantage of shear flow assays is that they offer a multitude of options to model healthy and diseased states.

#### 3.5.6. Immunoblotting and Co-Immunoprecipitation

As the field continues to push toward effective drugs for targeting β_2_-integrins, we must consider other proteins that may serve as therapeutic targets. Many of the assays already discussed can be adapted using target-specific inhibitors to probe the involvement of individual proteins in β_2_-integrin activation and function. Another useful method is immunoblotting, which continues to be a frequently utilized approach for determining specific cell signaling patterns. A significant portion of our understanding of integrin signaling comes from immunoblotting experiments. Through immunoblotting, key signaling molecules downstream of integrin activation, such as Syk, have been identified [148]. Specifically, Lefort et al. demonstrated that Mac-1 outside-in activation with ICAM-1 activates only the Akt apoptosis regulatory pathway and not the p38 MAPK pathway [108]. With immunoblotting, researchers can determine the signaling mechanism that underlies a given function [21]. 

Co-immunoprecipitation (Co-IP) assays are used to identify the protein–protein interactions occurring within cells by indirectly capturing proteins that are bound to specific target proteins [167]. The unknown proteins are then evaluated using traditional immunoblotting techniques. Co-IP has been useful for determining binding partners of β_2_-integrins in both neutrophils and lymphocytes [45,168]. Co-IP can also be used to determine CD18 binding partners on the surface of neutrophils [130]. Through Co-IP experiments, Willeke et al. demonstrated that Syk binds to CD18 in fibrinogen-stimulated neutrophils, expanding the understanding of Syk’s role in β_2_-integrin activation and signaling [66]. 

#### 3.5.7. Microscopy

Since the mid-1900s, researchers have utilized microscopy to evaluate neutrophils. The various methods of microscopy have aided researchers in their understanding of neutrophils, specifically how neutrophils change shape and polarize upon stimulation [164,169]. Confocal and immunofluorescence microscopy are the more commonly used microscopy platforms for evaluating neutrophils. Both methods utilize fluorescently labeled antibodies to evaluate β_2_-integrin clustering and surface distribution, subcellular localization, and colocalization with other proteins, such as F-actin [170,171,172,173]. 

Electron microscopy can be used to determine the binding of ligands to specific integrins. For example, Xu et al. used negative-stain electron microscopy to determine how the integrins α_M_β_2_ and α_X_β_2_ bind to iC3b, demonstrating that the different integrins bind to unique sites on iC3b [174]. Negative-stain electron microscopy can also be used to determine conformational changes of β_2_-integrins, deciphering between extended closed and extended open integrin conformations [135]. 

In a recent study, Wen et al. used high-resolution quantitative dynamic footprinting (qDF) microscopy, which is a total internal reflective fluorescence (TIRF)-based method, to analyze the relationship between kindlin-3 and β_2_-integrin activation in a shear flow assay of differentiated neutrophil-like HL60s [36]. Utilization of this method also demonstrated that integrins can obtain a high-affinity conformation (H^+^) without becoming extended (E^−^) while rolling along ICAM-1 [175]. This E^−^H^+^ conformation results in decreased neutrophil adhesion under flow. These findings were particularly interesting because they indicated an endogenous anti-inflammatory mechanism that could be harnessed by new integrin-targeting therapies [175]. 

### 3.6. In Vivo Experiments

While a vast amount of neutrophil β_2_-integrin research has been conducted in vitro, the use of mouse models has also made a significant impact on the field. The in vitro experiments utilizing primary cells or cell lines are essential for building a fundamental understanding of β_2_-integrins; however, in vitro findings are not always consistent with in vivo results. For example, in vitro experiments have consistently identified β_2_-integrins as essential receptors for neutrophil migration and adhesion, while more recent in vivo experiments have shown that the need for β_2_-integrins in vivo is variable. Table 5 summarizes a selection of in vivo models where β_2_-integrin dependence may vary depending on the stimulus or the organ in question. These differences are also noted when comparing leukocyte migration within a 3D collagen matrix versus placed on top of a collagen matrix [176,177]. The current understanding of these differences is that neutrophils are flexible in their responses to their environment. Migration on a 2D surface depends on cellular adhesion while movement within a 3D network, like collagen, depends on actomyosin contraction or actin polymerization [176,178,179,180]. What this does not explain is why pneumonia-causing pathogens have differential dependence on β_2_-integrins for neutrophil migration (Table 5) [181] or why certain β_2_-integrins are required while others are not [182]. These studies likely point toward evidence that the activation of internal cellular pathways following DAMP/PAMP recognition is also variable [183,184]. The recent development of humanized β_2_-integrin knockin mice should allow for better evaluation of integrin requirements in these disease models due to its ability to evaluate β_2_-integrin activation states in vivo [139].

#### Intravital Microscopy

Intravital microscopy in the mouse cremaster muscle is a well-established method for examining neutrophil function, characteristics, and interactions in the blood vessels. Leukocyte recruitment can be visualized in a variety of scenarios, including chemokine stimulation, fluorescently labeled leukocytes or transgenic mice expressing a fluorescent protein, and/or the application of pharmacological inhibitors [165,193,194]. Phillipson and colleagues used intravital microscopy to demonstrate the functional differences in LFA-1 and Mac-1 during MIP-2-induced neutrophil recruitment. Specifically, they found that LFA-1 is responsible for neutrophil adhesion while Mac-1 was responsible for neutrophil crawling [194]. This model was also used in experiments showing that E-selectin-induced slow rolling of neutrophils was LFA-1 dependent and Mac-1 independent [165]. As technology advances, researchers have expanded the field of intravital microscopy. Park and colleagues developed an intravital lung imaging system to examine neutrophil recruitment during sepsis-induced acute lung injury (ALI). In this study, investigators determined that decreased pulmonary microcirculation is due to obstructions of clustered neutrophils. Further, these neutrophils had high levels of surface Mac-1 expression, and the application of a Mac-1 inhibitor decreased sequestration in the pulmonary microvasculature [44]. Lim and colleagues also developed an advantageous model for evaluating β_2_-integrins. They generated a knockin mouse strain expressing CD11b conjugated to a monomeric yellow fluorescent protein (mYFP). This model can be utilized to image CD11b expressing cells in live mice or evaluate cell populations for CD11b expression in vitro. Although this approach results in all cells expressing CD11b to be YFP positive, this model allows for the analysis of functionally competent neutrophils in vivo without compromising β_2_-integrin–ligand interactions due to antibody binding [195]. Because of the breadth of options for examining neutrophils using intravital microscopy, this technique is an excellent next step for researchers wanting to translate in vitro findings into organ-specific in vivo scenarios [196].

## 4. Perspectives on the Study of Neutrophil β-Integrins

FcγRs often cooperate with β_2_-integrins in a complex mechanism of neutrophil activation and function. β_2_-integrin stimulation often results in neutrophil FcγR activation as well. For example, polyRGD is used to stimulate β_2_-integrins; however, neutrophils isolated from FcγR^−/−^ mice have diminished respiratory burst response to polyRGD stimulation, suggesting FcγR-cooperation with polyRGD activation and signaling [149]. Further, wild-type murine neutrophils stimulated with polyRGD displayed p38 MAPK phosphorylation, which is not activated when neutrophil β_2_-integrins, are stimulated in an outside-in manner using ICAM-1 [108,149]. Previous work by Jakus et al. demonstrated that stimulation of neutrophils with anti-integrin monoclonal antibodies requires both β_2_-integrins and FcγRs [122]. These findings are strengthened by the fact that many FcγR-stimulated neutrophil events are β_2_-integrin-dependent. For example, equine neutrophil adhesion and respiratory burst stimulated by low-density insoluble immune complexes are dependent on both FcγR and β_2_-integrins [123,124]. Other studies have also demonstrated the interconnectedness of FcγRs and β_2_-integrins [120,197]. These findings add a layer of difficulty to experiments designed for deciphering the intracellular signaling events exclusive to either FcγRs or β_2_-integrins.

Although this review has focused primarily on β_2_-integrins, neutrophils also express β_1_- and β_3_-integrins [7,198]. The majority of β-integrin research in neutrophils has focused on β_2_-integrins; however, these less-studied integrins may also play significant roles in neutrophil activation and signaling [14]. Lomakina and Waugh demonstrated that neutrophils significantly adhere to vascular cell adhesion molecule 1 (VCAM-1) through the integrin α_4_β_1_ [199]. Further, there is evidence to demonstrate that engagement of β_3_-integrins and β_1_-integrins activate β_2_-integrins [198,200,201]. While the presence of these “other” β-integrins adds additional complexity to investigations focused on β_2_-integrin-exclusive signaling, they also represent an opportunity for additional research that may lead to a more complete, and even potentially clinically relevant, understanding of neutrophil β-integrin receptor functions in vivo. 

## 5. Conclusions

β_2_-integrins have been the focus of intense research for decades, and many tools and assays have been developed to assess these receptors in neutrophils and neutrophil-like cells. These tools have been employed by researchers in very diverse fields, looking to decipher important questions regarding the biological function of β_2_-integrins. The combination of these cell types, tools, and assays offers a powerful resource to further understand β_2_-integrins and inform the future development of successful neutrophil targeting therapies.

## Figures and Tables

**Table 1 biomolecules-13-00892-t001:** Nomenclature for β_2_-integrins expressed on neutrophils.

β_2_-Integrin	Heterodimer	Other Names	Ligands
CD11a/CD18	α_L_β_2_	LFA-1	ICAM-1, ICAM-2, LPS
CD11b/CD18	α_M_β_2_	Mac-1, Complement receptor 3 (CR3)	iC3b, fibrinogen, factor X, ICAM-1, LPS
CD11c/CD18	α_X_β_2_	P150,95, Complement receptor 4 (CR4)	Fibrinogen, iC3b, collagen, ICAM-1, LPS, β-glucan
CD11d/CD18	α_D_β_2_		ICAM-3, VCAM-1, fibronectin, vitronectin, fibrinogen

**Table 2 biomolecules-13-00892-t002:** Diseases and disorders where neutrophil β_2_-integrins have been identified as key players in disease.

Disease or Disorder	Category	References
Antineutrophil cytoplasmic antibody (ANCA)-associated vasculitis (AAV)	Autoimmune disease	[19]
Aspergillus fumigatus	Fungal infections	[20,21]
Atrial fibrillation	Cardiovascular disease	[22]
Blood-brain barrier inflammation	Acute illness	[23,24]
Candida albicans	Fungal infections	[25]
COPD	Chronic disease	[26,27,28]
Interstitial lung disease (ILD)	Chronic inflammation, autoimmune disease	[29]
Ischemia-reperfusion injury	Acute injury/Sterile inflammation	[30,31,32,33,34]
Leukocyte adhesion deficiency (LAD)	Genetic disorder	[35,36,37]
Myocardial Infarction	Cardiovascular disease/Sterile inflammation	[33,34,38,39]
Rheumatoid arthritis	Autoimmune disease	[40]
SARS-CoV-2	Infectious disease	[41,42]
Sepsis	Acute illness	[43]
Sepsis-induced acute lung injury	Acute injury	[44,45]
Solid organ transplant rejection	Transplant rejection	[46]
Systemic lupus erythematosus (SLE)	Autoimmune disease	[12,40,47,48]
Thrombosis	Cardiovascular disease	[49]
Transfusion-related acute lung injury (TRALI)	Acute injury	[50]
Trauma/Vascular injury	Acute injury	[51,52,53]
Wiskott Aldrich syndrome	Genetic disorder	[54,55]

**Table 3 biomolecules-13-00892-t003:** Cell lines used in neutrophil β_2_-integrin research ^1^.

Cell Line	Requires Differentiation	Endogenous β_2_-Integrin Expression	Limitations/Drawbacks
HL60/PLB-985	Yes—DMSO or retinoic acid (referred to as dHL60s	α_L_β_2_, differentiation required for α_M_β_2_	Lack of specific and secretory granules, which limits upregulation of α_M_β_2_ following stimulation DMSO dHL60s have different IL-8R, signaling proteins, and α-actinin compared with humans. Lower α_M_β_2_ expression and dampened upregulation by fMLP and LTB4 compared with humans
K562	No	No	Requires stable transfection for α_M_β_2_ expression
HoxB8	Yes—GM-CSF or engraftment into mice	Yes—α_L_β_2_ and α_M_β_2_	Additional cytokines are needed to achieve “mature” neutrophilic cells. Lower levels of ROS production compared with murine neutrophils due to decrease gp91^phox^ expression. Lower chemotactic responses Not suitable for degranulation experiments

^1^ see the article text for references relevant to Table 3.

**Table 5 biomolecules-13-00892-t005:** Evidence for β_2_-integrin dependence in murine models in vivo.

Disease Model	β_2_-Integrin Dependent	References
Alzheimer’s disease	Yes	[185]
Atrial fibrosis	Yes	[22]
HMGB1-induced peritonitis	Mac-1: Yes LFA-1: No	[186]
Influenza	No	[187]
LTB_4_-induced intestinal transepithelial migration	Yes	[187]
Pneumonia		
*S. pneumonia*	Mac-1: yes LFA-1: no	[188] [181]
*P. aeruginosa*	Yes	[127,181]
*E. coli* LPS	Yes	[181]
Pulmonary aspergillosis	Yes	[20,189]
SLE-induced glomerular disease	Mac-1: No LFA-1: Yes	[190]
Thioglycollate peritonitis	Mac-1: No LFA-1: Yes	[186,191,192]

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
