# Peer review of "Targeting Neutrophil β2-Integrins: A Review of Relevant Resources, Tools, and Methods"

_biomolecules, 2023, doi:10.3390/biom13060892_

Round 1
Reviewer 1 Report
This manuscript summarizes and focuses on the relevant resources, research tools and scientific methods for the neutrophil Gr.[b]2-integrins. The contents are broad, the writing is comprehensive, and the language is fluent. A number of minor editing of the language may be needed. Here is the list (This list may not be thorough or complete).
Lines 2, 19, 65, 192, 530, 534: the "beta2-integrins" should be "Gr.[b]2-integrins" and the "beta2-integrin" should be "Gr.[b]2-integrin".
Line 21 and Line 96: "s" should be added to the word of "leukocyte" (please use the pl. form).
Line 26: delete Beta2 and the brackets. (only "Gr.[b]2-integrins" should be used as the subject of the sentence.
Lines 35, 36, 65(in Table 1), 293, 334, 530(in Table 4), 534 and 561: Please use "Gr.[b]2-integrins" instead of "B2-integrins" and use "Gr.[b]2-integrin" instead of "B2-integrin" (there is no such thing as B2 subfamily integrins).
Lines 50 and 589: Please use the lower case of "f" to replace the capitalized "F". The lower case of "f" should be used for N-formylmethionine-leucyl-phenylalanine.
Lines 59, 533 and 588, : please use the Greek letter "Gr.[g]" to replace the English spelling "gamma".
Lines 175 and 389: The "alpha" should be replaced by "Gr.[a]", and the "beta" should be replaced by "Gr.[b]".
Line 365: Please use "functional responses" to replace the word "functions".
Lines 293 and 593: Please use "intercellular" instead of "intracellular".
Line 519: The misspelling word "inhibditor".
The quality of English language is excellent.
Author Response
Response to Reviewer 1:
We would like to thank the reviewer for their careful review and feedback on this manuscript. We have used this feedback to make specific editorial changes (see below) and are pleased with the improvements.
Lines 2, 19, 65, 192, 530, 534: the "beta2-integrins" should be "Gr.[b]2-integrins" and the "beta2-integrin" should be "Gr.[b]2-integrin". – We have made this change throughout the manuscript and tables where appropriate.
Line 21 and Line 96: "s" should be added to the word of "leukocyte" (please use the pl. form). – We have made this change within the manuscript in lines 21 and 99.
Line 26: delete Beta2 and the brackets. (only "Gr.[b]2-integrins" should be used as the subject of the sentence. – We have made this change within the manuscript in line 26.
Lines 35, 36, 65(in Table 1), 293, 334, 530(in Table 4), 534 and 561: Please use "Gr.[b]2-integrins" instead of "B2-integrins" and use "Gr.[b]2-integrin" instead of "B2-integrin" (there is no such thing as B2 subfamily integrins). – We have made this change throughout the manuscript and tables as appropriate.
Lines 50 and 589: Please use the lower case of "f" to replace the capitalized "F". The lower case of "f" should be used for N-formylmethionine-leucyl-phenylalanine. – We have made this change in lines 51 and 636.
Lines 59, 533 and 588, : please use the Greek letter "Gr.[g]" to replace the English spelling "gamma". – We have made this change in lines 60 and 580. We have chosen to keep “gamma” spelled out in line 635 because it is a definition and its common occurrence in the literature as “Fc gamma receptor.”
Lines 175 and 389: The "alpha" should be replaced by "Gr.[a]", and the "beta" should be replaced by "Gr.[b]". – We have changed “alpha” to its Greek letter in line 187. All instances of “beta” have been changed throughout the manuscript to its Greek letter.
Line 365: Please use "functional responses" to replace the word "functions". – We have made this change in line 249 and 486.
Lines 293 and 593: Please use "intercellular" instead of "intracellular". – We have made this change in lines 327 and 640.
Line 519: The misspelling word "inhibditor". – We have corrected this misspelling in line 557.
Thank you again for your time invested in this review. We look forward to any additional feedback.
Reviewer 2 Report
In this review, the authors summarized our current understanding of neutrophil b2-integrin activation and function in health in diseases. They provided a complete overview of the tools, primary cells, cell lines, animal models, and current methods researchers use to study b2-integrin activation/function in vitro and validate that information in various mouse models (knock-out, knock-in, transgenic).
General: The review is well-written and informative. My only concern for a few issues is that the authors are softening the edges. Several statements should be more nuanced and consider differences in LFA-1 and Mac-1 subcellular localization in quiescent neutrophils. Furthermore, the authors could expand on limitations associated with cell lines and mouse neutrophils used in research. I invite the authors to consider the following points and clarification.
Page 1, lines 42-44. The sentence is too general and is not relevant for LFA-1. Mac-1 localizes in secondary (specific) and secretory granules. The increase in b2-integrin cell surface expression following PMN activation is mainly due to mobilization and granule fusion with the plasma membrane.
Page 4, last paragraph. It is correct saying that "quiescent" primary neutrophils are isolated from the mouse bone marrow or peripheral blood. The authors should include a statement highlighting that peripheral blood and bone marrow-derived neutrophils have slightly different phenotypes and behavior.
Page 4, last paragraph. Given that Fc gamma receptors can cooperate with b2-integrin in neutrophil activation, it is worthwhile mentioning the discrepancies in Fc gamma receptor expression between humans and mice. This is another limitation of using mouse neutrophils.
Tableau 3, paragraph 3.2. PLB-985 is also a human promyeloblast cell line currently used for understanding b2-integrin function and activation. I will also refer to this cell line that can be differentiated into neutrophil-like cells.
Another limitation of using HL60 cells (and PLB-895) is that differentiated cells lack specific and secretory granules. Is it an important point when studying Mac-1? Please add this point to Table 3 (Limitations/Drawbacks).
Section 3.3. Many antibodies are available for studying b2-integrin expression, activation/conformation states, and functions. It would include a table summarizing the common antibodies used to analyze their conformation (bent to extended conformation) and block b2-integrin in vitro and in vivo.
Page 12, line 519 (typo): inhibitor
Author Response
Response to Reviewer 2:
We would like to thank the reviewer for their thoughtful feedback on this manuscript. We have used this feedback to make the requested changes and are pleased with the improvements (see detailed improvements below). We have expanded the content to include a table detailing common anti-integrin antibodies and their purposes. We have also added discussion regarding nuances in β2-integrin biology and the limitations of neutrophil-like cell lines and murine neutrophils. These specific edits are documented as track changes in the revised manuscript submission.
Page 1, lines 42-44. The sentence is too general and is not relevant for LFA-1. Mac-1 localizes in secondary (specific) and secretory granules. The increase in b2-integrin cell surface expression following PMN activation is mainly due to mobilization and granule fusion with the plasma membrane. – We have added discussion regarding the nuances in β2-integrin biology throughout the paragraph between lines 37 and 65.
Page 4, last paragraph. It is correct saying that "quiescent" primary neutrophils are isolated from the mouse bone marrow or peripheral blood. The authors should include a statement highlighting that peripheral blood and bone marrow-derived neutrophils have slightly different phenotypes and behavior. – We have added extra discussion regarding murine neutrophils and their limitations in lines 167-174.
Page 4, last paragraph. Given that Fc gamma receptors can cooperate with b2-integrin in neutrophil activation, it is worthwhile mentioning the discrepancies in Fc gamma receptor expression between humans and mice. This is another limitation of using mouse neutrophils. – We have added a statement addressing murine neutrophil Fc gamma receptor expression in lines 167 and 168.
Tableau 3, paragraph 3.2. PLB-985 is also a human promyeloblast cell line currently used for understanding b2-integrin function and activation. I will also refer to this cell line that can be differentiated into neutrophil-like cells.
Another limitation of using HL60 cells (and PLB-895) is that differentiated cells lack specific and secretory granules. Is it an important point when studying Mac-1? Please add this point to Table 3 (Limitations/Drawbacks). – We have added PLB-985 cells to Table 3 and added the drawback regarding specific and secretory granules in Table 3. We have also added discussion on PLB-985 cells to the paragraph between line 216 and 229.
Section 3.3. Many antibodies are available for studying b2-integrin expression, activation/conformation states, and functions. It would include a table summarizing the common antibodies used to analyze their conformation (bent to extended conformation) and block b2-integrin in vitro and in vivo. – We have added a table detailing common anti-integrin antibodies and their purpose as Table 4.
Page 12, line 519 (typo): inhibitor – We have corrected this misspelling in line 557.
Thank you again for your time invested in this review. We look forward to any additional feedback you may have.